# Metabolomics Reveals the Effects of High Dietary Energy Density on the Metabolism of Transition Angus Cows

**DOI:** 10.3390/ani12091147

**Published:** 2022-04-29

**Authors:** Hao Chen, Chunjie Wang, Simujide Huasai, Aorigele Chen

**Affiliations:** 1College of Animal Science, Inner Mongolia Agricultural University, Hohhot 010018, China; chenhao9781@126.com (H.C.); smjd_2010@163.com (S.H.); 2College of Veterinary Medicine, Inner Mongolia Agricultural University, Hohhot 010018, China; chunjiewang200@sohu.com

**Keywords:** cow, dietary energy density, plasma, negative energy balance, metabolic profiles

## Abstract

**Simple Summary:**

The increase in the metabolic demand and the dramatically decreased feed intake of cows around parturition often cause a negative energy balance status in cows, which can cause metabolic disorders. Before parturition, dry matter intake of cows starts to decline, and this decline is practically unavoidable. Therefore, increasing the energy density of the diet is extremely important. We used untargeted metabolomics to reveal the effect of high dietary energy density on body metabolism and explore whether it can alleviate negative energy balance. Our research shows that feeding a high-energy diet could significantly improve antioxidant capacity, maintain phosphatidylcholine homeostasis and reduce the negative energy balance of cows by regulating lipid mobilization, muscle mobilization, and protein turnover.

**Abstract:**

The diet energy level plays a vital role in the energy balance of transition cows. We investigated the effects of high dietary energy density on body metabolism. Twenty multiparous Angus cows were randomly assigned to two treatment groups (10 cows/treatment), one receiving a high-energy (HE) diet (NEm = 1.67 Mcal/kg of DM) and the other administered a control (CON) diet (NEm = 1.53 Mcal/kg of DM). The results indicated that feeding a high-energy diet resulted in higher plasma glucose concentration and lower concentrations of plasma NEFA and BHBA on d 14 relative to calving in the HE-fed cows compared to the CON-fed ones. The postpartum plasma levels of T-AOC were lower in cows that received the CON diet than in cows in the HE group, while the concentration of malondialdehyde (MDA) showed an opposite trend. Among the 51 significantly different metabolites, the concentrations of most identified fatty acids decreased in HE cows. The concentrations of inosine, glutamine, and citric acid were higher in HE-fed cows than in CON-fed cows. Enrichment analysis revealed that linoleic acid metabolism, valine, leucine as well as isoleucine biosynthesis, and glycerophospholipid metabolism were significantly enriched in the two groups.

## 1. Introduction

During the transition period, the increase in the metabolic demand and the dramatically decreased feed intake of cows around parturition often cause a negative energy balance (NEB) status in cows, which can result in the mobilization of body fat reserves [1,2]. Extensive mobilization of body fat reserves around parturition and changes in metabolism cause metabolic disorders, such as fatty liver [3], displaced abomasum [4], and ketosis [5].

Dry matter intake (DMI) starts to decline before parturition, and this decline is practically unavoidable [6,7]. However, the energy requirement of the cow increases to support uterus growth in late pregnancy; therefore, increasing the energy density of the diet should facilitate the maintenance of energy intake [8,9]. Researchers have focused on alleviating the NEB status and fetal growth by increasing dietary energy density. Dry-rolled corn supplementation during late gestation could improve the energy status of the grazing cow [10]. Another research group reached a similar conclusion when comparing dairy cows fed a low-energy (1.58 Mcal/kg of DM) ration. Specifically, cows fed a high-energy diet (1.7 Mcal/kg DM) exhibited lower concentrations of plasma non-esterified fatty acids (NEFAs) on d 7 relative to calving [11,12] In recent years, studies have demonstrated that cows fed a pre-calving high-energy diet showed decreased concentrations of NEFA and β-hydroxybutyric acid (BHBA) in the prepartum period [13]. As the majority of the above studies were focused on the effects of increasing prepartum dietary energy density on the reproduction of cows, evidence is lacking assessing whether feeding high-energy diet during the entire transition period leads to comprehensive metabolic changes in beef cows. A growing number of studies have shown that overfeeding a high-energy diet during late gestation could affect DMI and the energy balance of postpartum cows, improving the health and liver function of the animals [14,15,16]. However, the comprehensive metabolic changes that occur as a result of increasing dietary energy density impact physiological and metabolic processes.

Metabolomics can quantitatively measure small molecular metabolites in biological samples using techniques such as nuclear magnetic resonance (NMR) [17], liquid chromatography –tandem mass spectrometry [18], and gas chromatography–mass spectrometry (GC–MS) [19]. Metabolomic technologies have been proposed as powerful tools for the early diagnosis of postpartum diseases, the identification of biomarkers, and metabolic pathway characterization in cattle [20,21,22]. Metabolomic technologies have also been used in transition cows to analyze the changes of metabolites in relation to nutrient levels and rumen-protected glucose supplementation [23,24]. LC–MS has been widely used in metabolomics studies due to its high detection sensitivity and the fact that it does require sample derivatization [25,26]. Furthermore, LC–MS is capable of detecting intact metabolites without chemical modification [27].

Therefore, we used untargeted metabolomics to reveal the effect of high dietary energy density on body metabolism and explore whether it can alleviate NEB. The metabolic changes in postpartum beef cows fed a high-energy diet examined from the perspective of small molecules were used to evaluate the effects of dietary energy.

## 2. Materials and Methods

The current study was performed in the Xilingol League, Inner Mongolia Autonomous Region, China. All procedures were approved and found to meet the animal welfare policies following the instructions of the China Council on Animal Care. The experimental procedures used were approved by the Institutional Animal Care and Use Committee of Inner Mongolia Agricultural University (Hohhot, China).

### 2.1. Animal and Experimental Design

Twenty multiparous Angus cows were selected from a larger breeding group and were randomly allocated into two groups. From day 45 before the expected day of calving to 35 days postpartum, each group was randomly assigned to two dietary energy levels (10 cows/treatment), and the cows in each energy treatment were assigned randomly to receive a high-energy (HE) diet (NEm = 1.67 Mcal/kg of DM) or a control (CON) diet (NEm = 1.53 Mcal/kg of DM), the diets was isonitrogenous (Table 1). The cows were offered the total mixed rations (TMR) twice daily (at 8 am and 3 pm) and given unlimited access to fresh water.

### 2.2. Collection and Analysis of Plasma

Blood samples (15 mL) were collected before morning feeding via jugular venipuncture 14 days after calving and then centrifuged at 3000 r/min for 10 min to harvest plasma. Plasma glucose, total protein, TG, insulin, leptin, and BUN concentrations were determined using commercially available bovine ELISA kits (Baoman Biological, Shanghai, China) according to the manufacturer’s protocols. Commercial colorimetric assay kits were used to measure the activities of glutathione peroxidase (GPx), superoxide dismutase (SOD), and catalase (CAT), as well as total antioxidant capacity (T-AOC), BHBA, and NEFA concentrations in the plasma.

### 2.3. Metabolite Extraction and LC–MS/MS Analysis

The metabolite extraction of all samples was according to Liu et al. [18]. The samples were then centrifuged at 13,000× *g* for 15 min at 4 °C, and 200 μL of the supernatant as transferred to a new vial for LC–MS analysis. Chromatographic separation was performed using an ACQUITY UPLC HSS T3 (100 mm × 2.1 mm i.d., 1.8 µm; Waters, Milford, CT, USA) preheated to 40 °C, as described by Gu et al. [28] and Yang et al. [29].

### 2.4. Statistical Analysis

Significance analyses of the body condition score (BCS), DMI, and plasma parameters in different groups were conducted using one-way ANOVA in SPSS. Significant differences are presented at the level of *p* < 0.05.

The raw data were first converted to CDF files by Thermo Scientific™ Xcalibur™, and the positive and negative data were imported into the SIMCA-P software package [30]. The retention time (RT) and MZ data were imported into the SIMCA-P software package for orthogonal projections to latent structures discriminate analysis (OPLS-DA) [31]. Using the Kyoto Encyclopedia of Genes and Genomes (KEGG, http://www.genome.jp/kegg, accessed on 20 April 2022), metabolic pathways mapped by every differential metabolite were acquired [30]. MetaboAnalyst (http://www.metaboanalyst.ca, accessed on 20 April 2022) was then used to analyze pathways [31], and the cutoff of the impact value from the topology analysis was set to 0.1 [32].

## 3. Results

### 3.1. Plasma Parameters

The main effects of transition energy density on the plasma parameters are summarized in Table 2. Our results indicated that feeding a high-energy diet resulted in higher plasma concentrations of glucose and insulin and in lower concentrations of plasma NEFA and BHBA on d 14 relative to calving compared with feeding the CON diet. Postpartum plasma levels of T-AOC were lower in cows that received the CON diet than in cows in the HE group, whereas the concentration of MDA showed an opposite trend. Furthermore, plasma SOD activity was similar between the treatments, while the CON diet tended to increase the GSH-PX levels.

### 3.2. Plasma Metabolomics Profiling

Figure 1 shows the overlap of the total ion chromatograms of the QC sample in the positive (Figure 1A) and negative (Figure 1B) ion modes. Score plots of the (O)PLS-DA are shown in Figure 2A,C and were performed to verify the different metabolites identified in the two groups. The R2Y values of the plasma (POS and NEG) were 0.993 and 0.990, respectively.

The permutation test results for the Q2 intercepts were −0.37 for serum POS and −0.43 for plasma NEG; the results are presented in Figure 2B,D. Both positive and negative data revealed a clear separation of the CON and HE groups. The significantly different metabolites were visualized through volcano plots (Figure 3), which clearly showed many different metabolites distinguishing the two groups.

As displayed in Table 3, among the 51 significantly different metabolites in the plasma, 22 metabolites had higher concentrations in the CON-fed cows than in the HE-fed cows (Table 3). Additionally, the concentrations of most identified fatty acids decreased in the HE-fed cows, including cibaric acid, linoleic acid, pelargonic acid, hexadecanedioic acid, heptadecanoic acid, and LysoPE (0:0/18:0). The concentrations of inosine, l-isoleucine, alpha-methylphenylalanine, glutamine, and citric acid were higher in the HE group.

The metabolome map revealed enriched pathways (*p* < 0.05) for metabolites that were identified in plasma (Figure 4). The enrichment analysis revealed that linoleic acid metabolism, valine, leucine, as well as isoleucine biosynthesis, and glycerophospholipid metabolism were significantly enriched in the HE-fed and CON-fed groups (Figure 4).

## 4. Discussion

Due to the decreasing DMI and rapidly increasing nutrient requirements after parturition, cows experience a period of NEB [33], which can be measured by NEFA and BHBA concentrations in transition cows [34]. Our results showed that cows in the HE groups exhibited higher concentrations of glucose and insulin and lower concentrations of plasma NEFA and BHBA on d 14 relative to calving compared with cows fed the CON diet. Our results are consistent with those of Sordillo and Raphael [35], who reported that increasing blood glucose results in higher insulin levels. Similar to our results, previous studies found that plasma concentrations of BHBA and NEFA in transition cows are lower when increasing dietary energy density [12,13]. Our data indicated that HE cows exhibited a lower degree of adipose tissue mobilization and that postpartum cows had a more positive energy balance. This finding may be due to higher ruminal glucogenic VFA production following the administration of the HE diet, which might improve hepatic glucose production and lead to lower fat mobilization in postpartum cows.

A balance exists between the levels of reactive oxygen species and those of endogenous antioxidants under normal cellular metabolism [36]. However, increased metabolic activity and lipid peroxidation are accompanied by reactive oxygen species production, and moreover, the overload of ROS may result in oxidative stress, which could induce tissue injury in transition dairy cows [37,38,39]. Superoxide dismutase, catalase, glutathione peroxidase, and malondialdehyde are well-known biomarkers of oxidative stress [40]. In this study, the HE-fed cows exhibited higher levels of T-AOC and lower MDA levels than did the CON-fed cows. This indicates that the degree of oxidative stress in the CON cows was enhanced further by the increased plasma NEFA concentrations. A recent study determined that hydrogen peroxide is produced as an initial metabolite, which could escalate ROS accumulation during the time of increased NEFA availability [41]. This may explain the increase in GSH-Px activities among the CON cows. As the role of GSH-Px is to convert hydrogen peroxide into harmless H_2_O, the increase in its content represents an increase in the content of hydrogen peroxide produced [42]. Moreover, in the present study, the reduction in BHBA levels paralleled the decrease in NEFA and MDA concentrations when administering the HE diet. Our results agree with other studies that reported that increased plasma markers of lipid peroxidation were correlated with NEFA concentration [43].

When nutrient intake cannot meet the energy requirements, lipids from the body reserves are mobilized, and fatty acids are released into the blood [44]. Contreras et al. [45] indicated that fatty acids circulate in various lipid fractions, such as NEFAs, neutral lipids, and phospholipids. In our study, the precursors related to the above fatty acids were downregulated by the HE diet, including LysoPE (0:0/18:0) and cibaric acid, which means that the HE diet reduced fat mobilization and lipid metabolism. While phosphatidylcholine plays an important role in triglyceride export by the liver [46], phosphatidylcholine is related to metabolic disease in cows [47]. However, LysoPE (0:0/18:0), a precursor of phosphatidylcholine, was downregulated in the HE group, which means that feeding a high-energy diet could maintain phosphatidylcholine homeostasis.

Lipid dystrophy and abnormal lipid metabolism could lead to insulin resistance and metabolic disorders [48]. Therefore, abnormal lipid metabolism results from lipid mediators, such as sphingomyelin, phosphatidylethanolamine, phosphatidylcholine, etc., which can induce oxidative stress and, in turn, trigger inflammatory reactions in lactating cows [49]. Plasma fatty acids, including linoleic acid, pelargonic acid, hexadecanoic acid, and cibaric acid, were decreased by the HE diet. These results are in accordance with the decreased NEFA and TG in the HE group. Some of the fatty acids detected at higher concentration in cows fed the HE diet could be of dietary origin, because the HE diet contained more “Unifat” and cotton seed oil, and dietary fatty acids could form part of the pool of available lipids with other catabolic sources of fatty acids [23]. Linoleic acid is a major n-6 PUFA that can be a substrate of phospholipaseA2 in the n-6 PUFA enzyme pathway [50]. Some studies found that n-6 PUFAs produce 13-hydroperoxyoctadecadienoic acid (13-HPODE), which, in turn, promotes the inflammatory response [51,52]. Furthermore, a low n-6 PUFA content may enhance immune functions [53], which possibly explains why the HE diet has positive effects on immunity and antioxidative capabilities. In addition, feeding an HE diet increased the levels of plasma inosine, which was proven to have a positive effect on the antioxidant and immunomodulatory capacity [54].

It is further speculated that the circulating levels of gluconeogenic amino acids play a role in the energy metabolism of beef cattle [55]. Specifically, the HE cows exhibited higher concentrations of threonine but lower concentrations of phenylalanine than the CON cows. In this study, despite feeding isonitrogenous diets, different feed sources were used containing different levels and concentrations of amino acids, which may have caused the differences in the circulating levels of gluconeogenic amino acids. While the gluconeogenic potential might have had a moderate effect on the energy status in Angus cows, the HE cows had greater concentrations of glutamine than the CON cows. This is relevant, as glutamine has been related to decreased catabolism of muscle in growing ruminants, and previous studies have indicated that glutamine could lead to protein synthesis and may act as an anabolic mediator promoting muscle growth [56,57]. Furthermore, some gluconeogenic amino acids were associated with residual feed intake in black Angus beef steers, which may explain the differences in dry matter intakes between the two groups [58]. It has also been determined that hydroxyproline arises biosynthetically from proline through an enzymatic catalytic reaction with hydroxylase [59]. Along this line, Mata [60] indicated that increased plasma hydroxyproline concentrations are linked to alcoholic liver cirrhosis. In this study, the hydroxyproline concentration was lower in HE group, which could be due to the higher gluconeogenesis and greater muscle protein catabolism in the CON cows.

Some branched-chain amino acids (BCAAs) could be used for gluconeogenesis, whereas the presence of antagonists among BCAAs may lead to opposite changes in l-isoleucine [61,62]. However, this conjecture requires further investigation. Creatine plays a vital role in energy metabolism [63]. For example, Luo et al. [64] found that higher concentrations of plasma creatine were detected in dairy cows with severe NEB, while a previous study speculated that a higher concentration of creatine may indicate extensive mobilization of phosphocreatine in the muscle tissue to supply energy [65]. In recent years, researchers found a negative correlation between plasma creatine and the feed efficiency of beef steers [55,62]. Thus, the lower concentration of creatine in HE cows may be a signal of less muscle mobilization and reduced protein turnover.

## 5. Conclusions

In summary, feeding an HE diet decreased the concentrations of NEFA and BHBA, increased T-AOC concentrations, lowered the MDA levels, and lowered GSH-Px activity in postpartum cows when compared with CON cows. Furthermore, the GC–TOF/MS analysis of plasma indicated that feeding an HE diet reduced lipid metabolism and downregulated various metabolites, including LysoPE (0:0/18:0), linoleic acid, and cibaric acid, which means that feeding a high-energy diet could maintain phosphatidylcholine homeostasis. More specifically, the HE cows exhibited higher concentrations of threonine and glutamine but lower concentrations of phenylalanine than the CON cows. Therefore, feeding an HE diet could reduce the NEB of cows by regulating lipid mobilization, muscle mobilization, and protein turnover. Accordingly, these results provide deeper insight into the molecular mechanism by which an HE diet may influence the energy status of cows.

## Figures and Tables

**Figure 1 animals-12-01147-f001:**
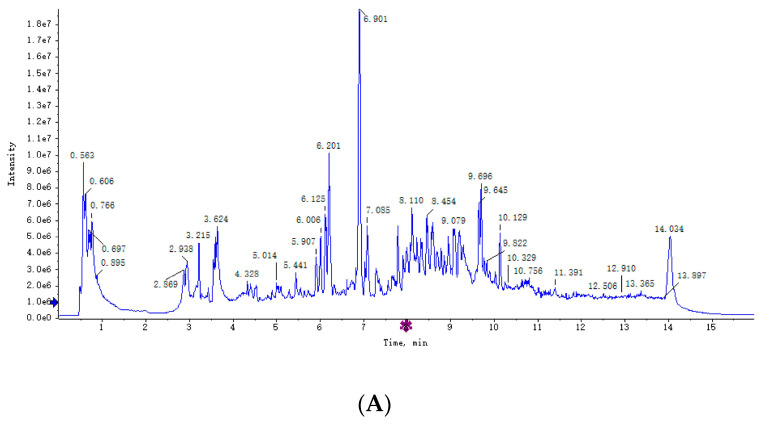
Overlap of the total ion chromatograms of the QC sample in the positive (**A**) and negative (**B**) ion modes.

**Figure 2 animals-12-01147-f002:**
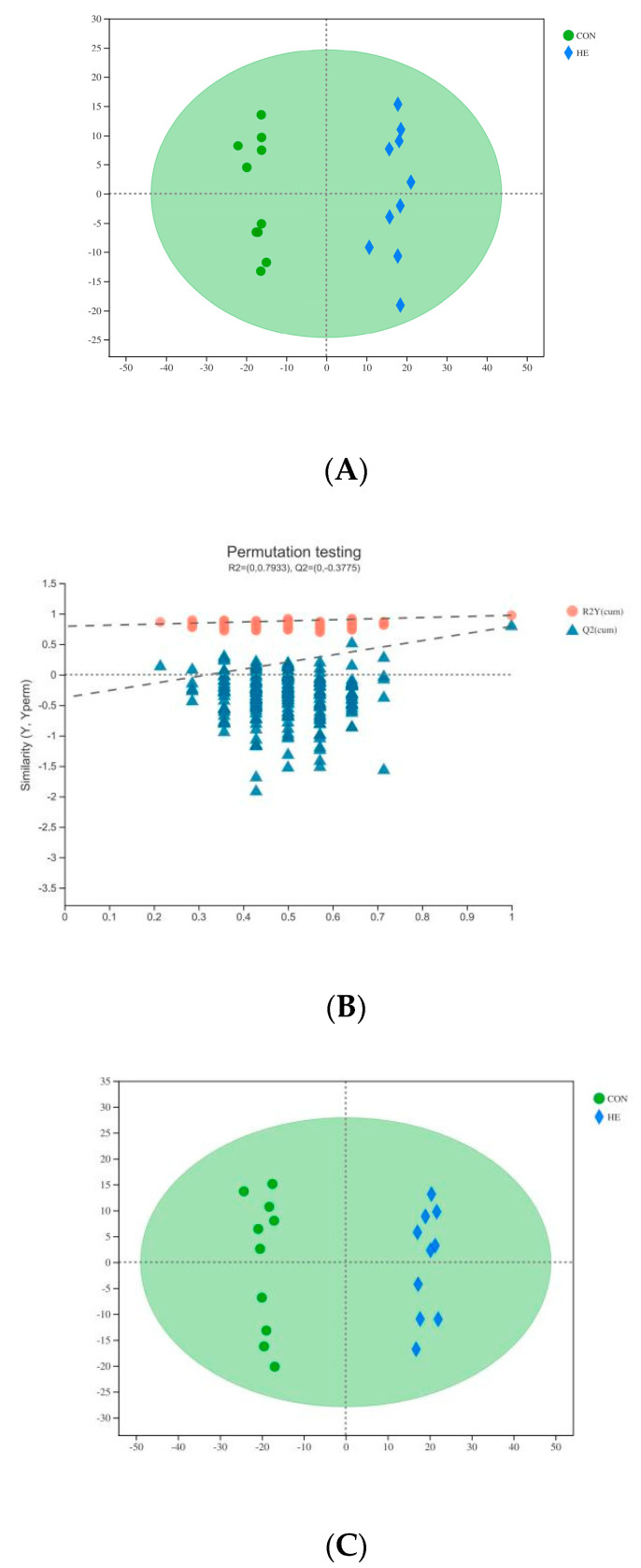
The OPLS-DA model (**A**,**C**) and permutation test of the OPLS-DA model (**B**,**D**) were derived from the LC–MS metabolomics profiles of the plasma of Angus cows fed the high-energy diet (HE) and the control diet (CON). (**A**–**D**) plasma samples analyzed in POS and NEG ion modes; the green and blue colors indicate, respectively, CON and HE diets administered to beef steers (**A**,**C**). Blue triangles represent the Q2 value, and red dots represent the R2 value from the permutation test (**B**,**D**).

**Figure 3 animals-12-01147-f003:**
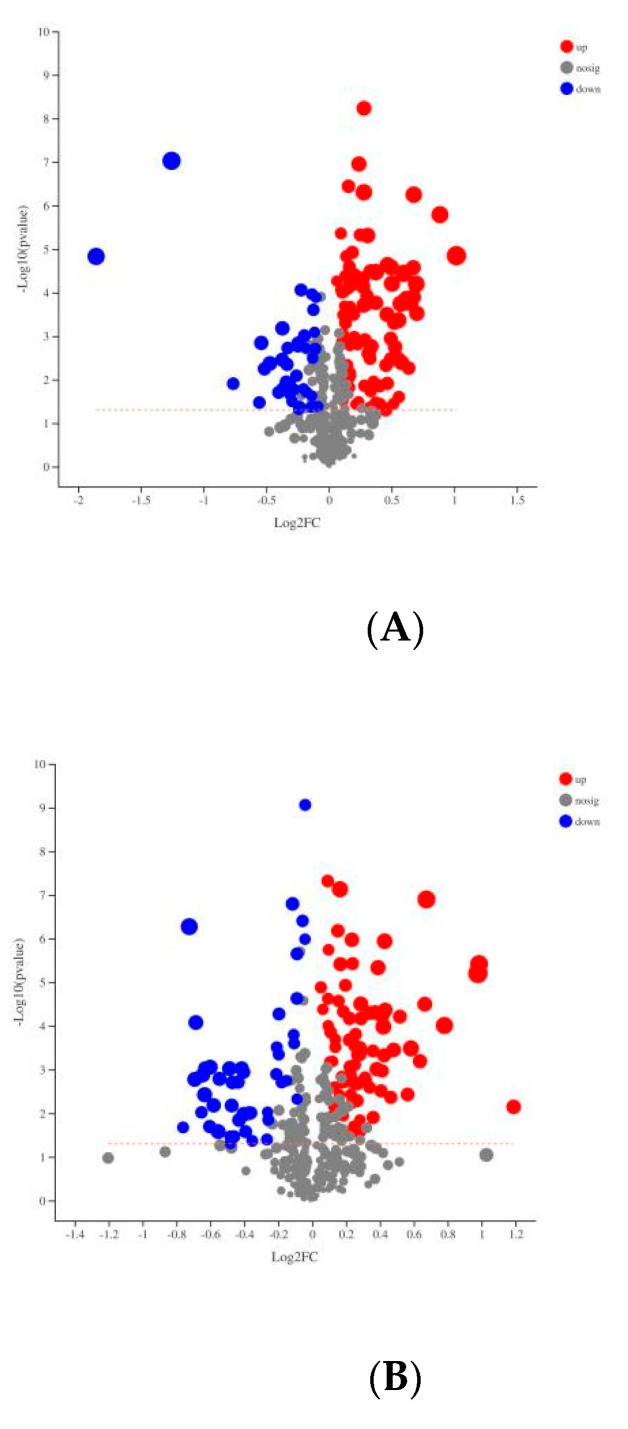
Volcano plots for the HE and. CON groups. (**A**,**B**) plasma samples analyzed in POS and NEG ion modes, respectively. Red and blue indicate, respectively, significantly upregulated and downregulated metabolites in the HE group compared with the CON group, and gray denotes no significant difference.

**Figure 4 animals-12-01147-f004:**
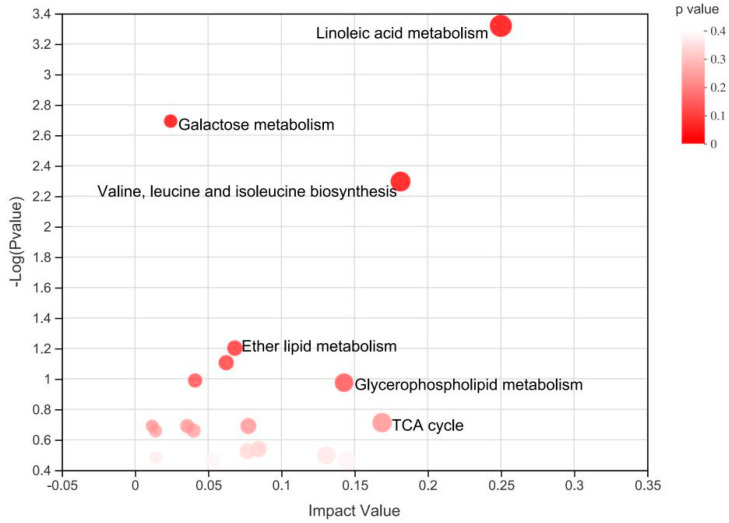
Metabolome map of common metabolites identified in plasma. The *x*-axis represents the pathway impact, and the *y*-axis represents the pathway enrichment.

**Table 1 animals-12-01147-t001:** Chemical composition of the mixed diets (HE and CON) offered to the cows during the trial.

		Group ^1^
Item	HE	CON
Ingredient, % of DM		
Corn silage	42.0	33.0
Dry rice straw	18.0	27.0
Corn	22.8	15.6
Wheat bran	5.2	12.1
Soybean meal	2.4	2.4
Cottonseed meal	3.6	5.1
CaHPO_4_	0.6	0.6
NaHCO_3_	0.4	0.4
NaCl	0.5	0.5
Unifat ^2^	2.5	1.3
Premix ^3^	2.0	2.0
Total	100	100
Nutrient composition		
CP, % of DM	11.55	11.78
ADF, % of DM	23.52	26.88
NDF, % of DM	37.51	41.62
TDN, % of DM	71.19	68.31
NEm ^4^, Mcal/kg DM	1.67	1.53
Calcium, % of DM	0.72	0.79
Phosphorus, % of DM	0.36	0.34

^1^ HE = high energy (NEm = 1.67 Mcal/kg of DM); CON = medium energy; (NEm = 1.53 Mcal/kg of DM); ^2^ Fractionated palm fatty acids (China Benefit Agriculture, Beijing, China). ^3^ Premix contained (per kg of premix): 480,000 IU of vitamin A, 90,000 IU of vitamin D_3_, 3500 IU of vitamin E, 2400 mg of Fe, 168 g of Ca, 38 g of P, 950 mg of Cu, 1500 mg of Mn, 3150 mg of Zn, 28 mg of I, 33 mg of Se, and 26 mg of Co. ^4^ NEm was calculated based on NRC (2001).

**Table 2 animals-12-01147-t002:** Effect of dietary energy level during the transition period on the concentration of serum metabolites and the oxidative status in postpartum Angus cows.

		Treatments ^1^		
Items ^2^	CON	HE	SEM	*p*-Value
Glucose mmol/L	5.73 ^b^	6.21 ^a^	0.055	0.031
Insulin mIU/L	19.02 ^b^	28.47 ^a^	0.398	0.002
NEFA μmol/L	612.56 ^a^	529.77 ^b^	30.50	0.017
BHBA μmol/L	509.62 ^a^	398.31 ^b^	18.69	0.026
T-AOC U/mL	2.25 ^b^	3.63 ^a^	0.375	<0.001
GSH-Px U/mL	136.05	102.03	16.53	0.011
SOD U/mL	15.66	17.91	0.021	0.127
MDA mmol/mL	2.33 ^a^	1.56 ^b^	0.76	0.018

^1^ HE = high energy (NEm = 1.67 Mcal/kg of DM); ^2^ CON = medium energy; (NEm = 1.53 Mcal/kg of DM); ^a,b^ Means bearing different superscripts in the same row differ significantly (*p* < 0.05).

**Table 3 animals-12-01147-t003:** Identification of different serum metabolites in cows fed the HE diet compared to cows in the control group.

Metabolite Name	VIP	RT (min)	Ion (m/z)	Fold Change	*p*-Value	Positive/Negative
Cibaric acid	1.2626	6.67	325.20	0.84	0.0378	pos
Linoleic acid	2.2307	7.78	485.32	0.78	<0.00001	pos
Pelargonic acid	1.6576	6.51	200.16	0.72	0.045	pos
2-Hydroxymyristic acid	1.0692	8.41	243.19	1.1034	0.0061	neg
Citric acid	1.235	0.727	191.02	1.06	0.0005	neg
Hexadecanedioic acid	1.0971	7.63	285.21	0.93	<0.00001	neg
heptadecanoic acid	1.1185	6.13	263.24	0.61	0.0015	neg
7-Ketodeoxycholic acid	1.9979	6.97	451.36	1.57	0.00018	neg
11Z-Eicosenoic acid	1.0659	9.16	355.28	1.065	0.0014	neg
Butyl salicylate	1.033	6.66	195.10	0.9458	0.02184	pos
LysoPE(0:0/22:0)	1.1191	9.34	582.37	0.94	0.0005	neg
LysoPC(20:4(5Z,8Z,11Z,14Z))	1.2623	7.96	544.33	0.95	0.01638	pos
LysoPE (0:0/18:0)	1.1836	7.98	550.31	0.91	0.0006	neg
LysoPE(0:0/22:4(7Z,10Z,13Z,16Z))	1.3637	7.85	574.31	0.79	0.0033	neg
LysoPC(20:0)	1.3302	9.20	552.40	0.94	0.00001	pos
LysoPC(22:2(13Z,16Z))	1.4444	9.27	620.39	0.83	0.0002	neg
LysoPC(O-18:0)	1.8298	9.35	544.36	0.67	0.0004	neg
PS(14:0/18:1(9Z))	2.41	9.16	778.48	0.46	0.0002	neg
l-Phosphoarginine	2.3631	3.53	237.07	0.48	0.02184	pos
Alpha-methylphenylalanine	1.8553	2.15	180.10	1.74	0.022	pos
Xanthosine	1.6569	5.02	319.04	0.6661	0.03052	neg
Pilosine	1.5464	5.32	287.15	0.7331	0.024	pos
l-Pyridosine	1.85	2.74	237.12	1.91	0.011	pos
Avenanthramide L	1.9264	3.35	348.08	0.6458	0.0375	pos
Shinflavanone	1.6465	7.01	411.15	1.3989	0.00027	neg
Mytilin B	1.6406	4.11	391.10	0.7846	0.0403	pos
3-phenyllactic acid	1.540	4.14	131.05	0.8978	0.0001	pos
Threonine	1.667	5.36	310.12	1.274	0.01084	neg
6-Hydroxymelatonin	1.5301	5.42	293.11	1.4486	0.0009	neg
Inosine	1.5043	2.33	305.02	1.3865	0.02384	neg
Glutamine	1.8506	3.09	292.13	1.67	0.00157	pos
Galactose	1.1852	11.3	191.02	1.17	0.0011	neg
Butyramide	1.1267	1.56	88.07	0.8722	0.03654	pos
Physapubenolide	1.101	6.13	563.24	0.91	0.0015	neg
l-Isoleucine	1.088	5.13	132.10	1.053	0.0375	pos
l-ornithine	1.072	6.641	173.09	1.089	0.0004	neg
l-valine	1.176	5.82	165.55	0.741	0.001	neg
l-hydroxyproline	1.2108	2.71	277.12	0.847	0.0149	neg
Isopentenyladenine-9-*N*-glucoside	1.0246	7.83	405.22	1.0745	0.0078	pos
Gamma-Glutamylvaline	1.7916	1.78	245.11	1.68	<0.00001	neg
Deoxycholic acid	1.74	7.79	437.28	1.12	<0.00001	neg
5′-Deoxy-5-fluorocytidine	1.60	5.07	268.0	0.72	0.008	pos
Acetoxy-8-gingerol	1.53	6.79	363.21	1.23	0.0003	neg
Gamma-Glu-Leu	1.512	2.86	259.12	1.32	0.01	neg
Oleyl alcohol	1.135	7.58	286.30	0.81	0.048	pos
4-Methoxybenzyl propanoate	1.1245	5.26	195.10	1.21	0.017	pos
Sphinganine	1.062	6.08	274.27	0.91	0.034	pos
Lucidenic acid E2	1.042	5.77	551.24	1.14	0.0039	neg
Monomenthyl succinate	1.26	4.60	301.16	1.25	0.001	neg
Cinncassiol D1 glucoside	1.23	5.74	535.25	1.22	0.006	neg
*N*-Decanoylglycine	1.16	6.65	228.16	1.35	0.04	neg

VIP: variable importance in projection; RT: retention time; fold change: HE group vs. CON group.

## Data Availability

The datasets that is applied and/or analyzed throughout the prevailing research are available from the corresponding writer upon fair appeal.

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
