# Peer review of "Metabolomics Reveals the Effects of High Dietary Energy Density on the Metabolism of Transition Angus Cows"

_animals, 2022, doi:10.3390/ani12091147_

Round 1

Reviewer 1 Report

animals-1694588-peer-review-v1

The manuscript: „Metabolomics reveals the effects of high dietary energy density 1 on the metabolism of transition Angus cows” prepared by Hao Chen et al. is very interesting and quait well presented. Authors investigated the effects of high dietary energy density on body metabolism. Twenty multiparous Angus cows were randomly assigned to two treatment condition (ten cows/treatment), and the cows in each treatment were assigned randomly to receive a high-energy (HE) density diet (NEm =1.67 12 Mcal/kg of DM) or control (CON) diet (NEm =1.53 Mcal/kg of DM). The results indicated that feeding a high-energy diet results in higher glucose concentration and lower concentrations of plasma NEFA and BHBA on d 14 relative to calving. Postpartum plasma levels of T-AOC were lower in cows that received the CON diet than in cows in the HE group, and the concentration of MDA was opposite that of T-AOC. Among the 51 significantly differential metabolite, the concentrations of most identified fatty acids decreased in the HE cows, The levels of inosine, glutamine, and citric acid were higher in the HE-fed cows than in the CON cows. Enrichment analysis reveals that linoleic acid metabolism, valine, leucine as well as isoleucine biosynthesis and glycerophospholipid metabolism were significantly enriched between the two groups.

Details:

Line 78   pleasure change to  8 am and 3 pm

Line 101 please remove one “were”

Line 141- 142:  Figure 2. Orthogonal partial least squares discriminant analysis ((O)PLS-DA) plot of cow plasma metabolites in comparisons of HE and CON following (A,B) positive and (C,D) negative mode ionization.

Poorly described graph (A and C in the legend have HE and CON.

Line 167 Table 3 should be included in the chapter “2.1. Animal and Experimental Design”

Line 248  pleasure change to  Mata [60] indicated that increased plasma hydroxyproline levels are linked to alcoholic liver cirrhosis.

Line 254-255  pleasure change to   Luo et al. [64] found that higher concentrations of plasma creatine were detected in dairy cows with severe NEB, while a previous study speculated that a higher concentration of creatine may indicate extensive mobilization of phosphocreatine in the muscle tissue to supply energy[65].

Reviewer 2 Report

This is an interesting paper that adds useful information about the effects of negative energy balance to beef cow nutrition and management. It is recommended that the authors refine the description of the experimental design - see comments in lines 75 to 77. Simplify by referring to "dietary energy levels", rather than treatment condition. The authors tend to get confused about the use of levels vs concentrations. It is acceptable to refer to specific energy "levels", but when referring to specific nutrients or metabolistes which were measured, rather use "concentrations". 

It is also recommended that the description of the nutrient composition of diets (Table 3) be moved closed to where it is described first, e.g. in Materials and methods section.

In the discussion of the results the authors should explain more clearly that several of the "metabolites" and in particular fatty acids and amino acids could originate from either dietary or endogenous origin. This complicates assumptions about the metabolism of some of those metabolites between the dietary energy treatment groups in this study. For example,  some of the fatty acids detected at higher concentration in cows fed the HE-diet may be of dietary origin, because the HE diet contained double the amounts of "Unifat", more cotton seed oil meal, wheat bran, dry rice straw and corn silage. It should be explained here that dietary fatty acids may contribute to the available pool of fatty acids, bypass rumen biohydrogenation and form part of the pool of available lipids with other catabolic sources of fatty acids or from lipolysis (ketones).  

In line 250, it is unlikely that liver function will be compromised in CON cows, since they did not experience severe nutrition stress. The high hydroxyproline was more likely due to the higher gluconeugenesis in the CON cows and greater muscle protein catabolism, which may explain the increase in hydroxyproline?
